# Galileo: Perceiving Physical Object Properties by Integrating a Physics Engine with Deep Learning

**Jiajun Wu**[*]
EECS, MIT
jiajunwu@mit.edu

**Ilker Yildirim**[*]
BCS MIT, The Rockefeller University
ilkery@mit.edu

**Joseph J. Lim**
EECS, MIT
lim@csail.mit.edu

**William T. Freeman**
EECS, MIT
billf@mit.edu

**Joshua B. Tenenbaum**
BCS, MIT
jbt@mit.edu

## Abstract

Humans demonstrate remarkable abilities to predict physical events in dynamic scenes, and to infer the physical properties of objects from static images. We propose a generative model for solving these problems of physical scene understanding from real-world videos and images. At the core of our generative model is a 3D physics engine, operating on an object-based representation of physical properties, including mass, position, 3D shape, and friction. We can infer these latent properties using relatively brief runs of MCMC, which drive simulations in the physics engine to fit key features of visual observations. We further explore directly mapping visual inputs to physical properties, inverting a part of the generative process using deep learning. We name our model Galileo, and evaluate it on a video dataset with simple yet physically rich scenarios. Results show that Galileo is able to infer the physical properties of objects and predict the outcome of a variety of physical events, with an accuracy comparable to human subjects. Our study points towards an account of human vision with generative physical knowledge at its core, and various recognition models as helpers leading to efficient inference.

## 1 Introduction

Our visual system is designed to perceive a physical world that is full of dynamic content. Consider yourself watching a Rube Goldberg machine unfold: as the kinetic energy moves through the machine, you may see objects sliding down ramps, colliding with each other, rolling, entering other objects, falling — many kinds of physical interactions between objects of different masses, materials and other physical properties. How does our visual system recover so much content from the dynamic physical world? What is the role of experience in interpreting a novel dynamical scene?

Recent behavioral and computational studies of human physical scene understanding push forward an account that people's judgments are best explained as probabilistic simulations of a realistic, but mental, physics engine [2, 8]. Specifically, these studies suggest that the brain carries detailed but noisy knowledge of the physical attributes of objects and the laws of physical interactions between objects (*i.e.*, Newtonian mechanics). To understand a physical scene, and more crucially, to predict the future dynamical evolution of a scene, the brain relies on simulations from this mental physics engine. Even though the probabilistic simulation account is very appealing, there are missing practical and conceptual leaps. First, as a practical matter, the probabilistic simulation approach is shown to work only with synthetically generated stimuli: either in 2D worlds, or in 3D worlds but each

---

[*] Indicates equal contribution. The authors are listed in the alphabetical order.

object is constrained to be a block and the joint inference of the mass and friction coefficient is not handled [2]. Second, as a conceptual matter, previous research rarely clarifies how a mental physics engine could take advantage of previous experience of the agent [11]. It is the case that humans have a life long experience with dynamical scenes, and a fuller account of human physical scene understanding should address it.

Here, we build on the idea that humans utilize a realistic physics engine as part of a generative model to interpret real-world physical scenes. We name our model Galileo. The first component of our generative model is the physical object representations, where each object is a rigid body and represented not only by its 3D geometric shape (or volume) and its position in space, but also by its mass and its friction. All of these object attributes are treated as latent variables in the model, and are approximated or estimated on the basis of the visual input.

The second part is a fully-fledged realistic physics engine — in this paper, specifically the Bullet physics engine [4]. The physics engine takes a scene setup as input (*e.g.*, specification of each of the physical objects in the scene, which constitutes a hypothesis in our generative model), and physically simulates it forward in time, generating simulated velocity profiles and positions for each object.

The third part of Galileo is the likelihood function. We evaluate the observed real-world videos with respect to the model's hypotheses using the velocity vectors of objects in the scene. We use a standard tracking algorithm to map the videos to the velocity space.

Now, given a video as observation to the model, physical scene understanding in the model corresponds to inverting the generative model by probabilistic inference to recover the underlying physical object properties in the scene. Here, we build a video dataset to evaluate our model and humans on real-world data, which contains 150 videos of different objects with a range of materials and masses over a simple yet physically rich scenario: an object sliding down an inclined surface, and potentially collide with another object on the ground. Note that in the fields of computer vision and robotics, there have been studies on predicting physical interactions or inferring 3D properties of objects for various purposes including 3D reasoning [6, 13] and tracking [9]. However, none of them focused on learning physical properties directly, and nor they have incorporated a physics engine with representation learning.

Based on the estimates we derived from visual input with a physics engine, a natural extension is to generate or synthesize training data for any automatic learning systems by bootstrapping from the videos already collected, and labeling them with estimates of Galileo. This is a self-supervised learning algorithm for inferring generic physical properties, and relates to the wake/sleep phases in Helmholtz machines [5], and to the cognitive development of infants. Extensive studies suggest that infants either are born with or can learn quickly physical knowledge about objects when they are very young, even before they acquire more advanced high-level knowledge like semantic categories of objects [3, 1]. Young babies are sensitive to physics of objects mainly from the motion of foreground objects from background [1]; in other words, they learn by watching *videos* of moving objects. But later in life, and clearly in adulthood, we can perceive physical attributes in just static scenes without any motion.

Here, building upon the idea of Helmholtz machiness [5], our approach suggests one potential computational path to the development of the ability to perceive physical content in static scenes. Following the recent work [12], we train a recognition model (*i.e.*, sleep cycle) that is in the form of a deep convolutional network, where the training data is generated in a self-supervised manner by the generative model itself (*i.e.*, wake cycle: real-world videos observed by our model and the resulting physical inferences). Interestingly, this computational solution asserts that the infant starts with a relatively reliable mental physics engine, or acquires it soon after birth.

Our work makes three contributions. First, we propose Galileo, a novel model for estimating physical properties of objects from visual inputs by incorporating the feedback of a physics engine in the loop. We demonstrate that it achieves encouraging performance on a real-world video dataset. Second, we train a deep learning based recognition model that leads to efficient inference in the generative model, and enables the generative model to predict future dynamical evolution of static scenes (*e.g.*, how would that scene unfold in time). Third, we test our model and compare it to humans on a variety of physical judgment tasks. Our results indicate that humans are quite successful in these tasks, and our model closely matches humans in performance, but also consistently makes

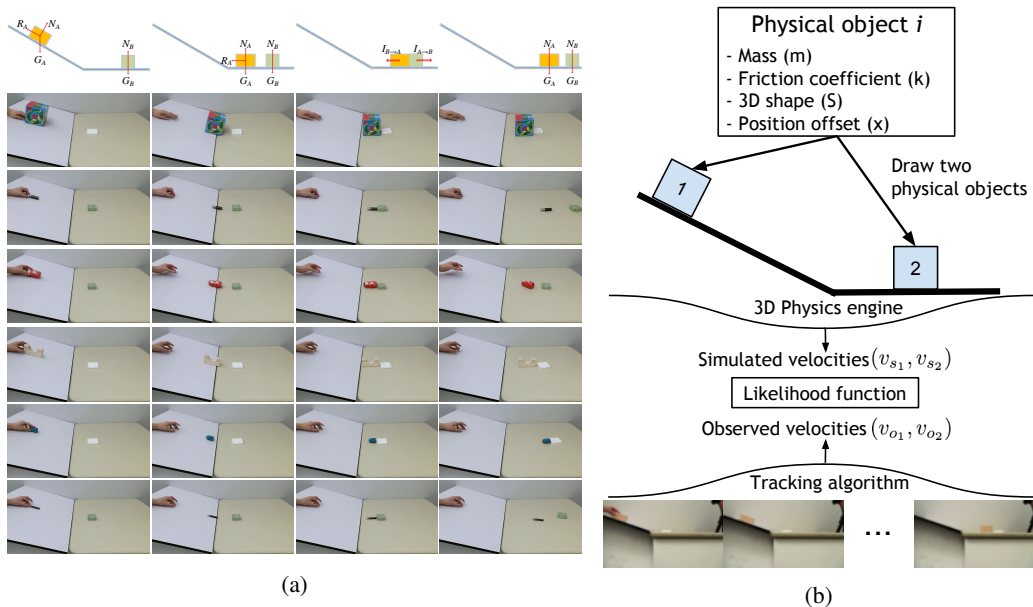

(a)                                                         (b)

Figure 1: (a) Snapshots of the dataset. (b) Overview of the model. Our model formalizes a hypothesis space of physical object representations, where each object is defined by its mass, friction coefficient, 3D shape, and a positional offset w.r.t. an origin. To model videos, we draw exactly two objects from that hypothesis space into the physics engine. The simulations from the physics engine are compared to observations in the velocity space, a much "nicer" space than pixels.

similar errors as humans do, providing further evidence in favor of the probabilistic simulation account of human physical scene understanding.

## 2 Scenario

We seek to learn physical properties of objects by observing videos. Among many scenarios, we consider an introductory setup: an object is put on an inclined surface; it may either slide down or keep static due to gravity and friction, and may hit another object if it slides down.

This seemingly simple scenario is physically highly involved. The observed outcome of these scenario are physical values which help to describe the scenario, such as the velocity and moving distance of objects. Causally underlying these observations are the latent physical properties of objects such as the material, density, mass and friction coefficient. As shown in Section 3, our Galileo model intends to model the causal generative relationship between these observed and unobserved variables.

We collect a real-world video dataset of about 100 objects sliding down a ramp, possibly hitting another object. Figure 1a provides some exemplar videos in the dataset. The results of collisions, including whether it will happen or not, are determined by multiple factors, such as material (density and friction coefficient), size and shape (volume), and slope of surface (gravity). Videos in our dataset vary in all these parameters.

Specifically, there are 15 different materials — cardboard, dough, foam, hollow rubber, hollow wood, metal coin, metal pole, plastic block, plastic doll, plastic ring, plastic toy, porcelain, rubber, wooden block, and wooden pole. For each material, there are 4 to 12 objects of different sizes and shapes. The angle between the inclined surface and the ground is either $10^o$ or $20^o$. When an object slides down, it may hit either a cardboard box, or a piece of foam, or neither.

# 3 Galileo: A Physical Object Model

The gist of our model can be summarized as probabilistically inverting a physics engine in order to recover unobserved physical properties of objects. We collectively refer to the unobserved latent variables of an object as its *physical representation $T$*. For each object $i$, $T_i$ consists of its mass $m_i$, friction coefficient $k_i$, 3D shape $V_i$, and position offset $p_i$ w.r.t. an origin in 3D space.

We place uniform priors over the mass and the friction coefficient for each object: $m_i \sim \text{Uniform}(0.001, 1)$ and $k_i \sim \text{Uniform}(0, 1)$, respectively.

For 3D shape $V_i$, we have four variables: a shape type $t_i$, and the scaling factors for three dimensions $x_i, y_i, z_i$. We simplify the possible shape space in our model by constraining each shape type $t_i$ to be one of the three with equal probability: a box, a cylinder, and a torus. Note that applying scaling differently on each dimension to these three basic shapes results in a large space of shapes.[1] The scaling factors are chosen to be uniform over the range of values to capture the extent of different shapes in the dataset.

Remember that our scenario consists of an object on the ramp and another on the ground. The position offset, $p_i$, for each object is uniform over the set $\{0, \pm 1, \pm 2, \cdots, \pm 5\}$. This indicates that for the object on the ramp, its position can be perturbed along the ramp (*i.e.*, in 2D) at most 5 units upwards or downwards from its starting position, which is 30 units upwards on the ramp from the ground.

The next component of our generative model is a fully-fledged realistic physics engine that we denote as $\rho$. Specifically we use the Bullet physics engine [4] following the earlier related work. The physics engine takes a specification of each of the physical objects in the scene within the basic ramp setting as input, and simulates it forward in time, generating simulated velocity vectors for each object in the scene, $v_{s_1}$ and $v_{s_2}$ respectively – among other physical properties such as position, rendered image of each simulation step, *etc*.

In light of initial qualitative analysis, we use velocity vectors as our feature representation in evaluating the hypothesis generated by the model against data. We employ a standard tracking algorithm (KLT point tracker [10]) to "lift" the visual observations to the velocity space. That is, for each video, we first run the tracking algorithm, and we obtain velocities by simply using the center locations of each of the tracked moving objects between frames. This gives us the velocity vectors for the object on the ramp and the object on the ground, $v_{o_1}$ and $v_{o_2}$, respectively.

Given a pair of observed velocity vectors, $v_{o_1}$ and $v_{o_2}$, the recovery of the physical object representations $T_1$ and $T_2$ for the two objects via physics-based simulation can be formalized as:

$$P(T_1, T_2 | v_{o_1}, v_{o_2}, \rho(\cdot)) \propto P(v_{o_1}, v_{o_2} | v_{s_1}, v_{s_2}) \cdot P(v_{s_1}, v_{s_2} | T_1, T_2, \rho(\cdot)) \cdot P(T_1, T_2). \quad (1)$$

where we define the likelihood function as $P(v_{o_1}, v_{o_2} | v_{s_1}, v_{s_2}) = N(v_o | v_s, \Sigma)$, where $v_o$ is the concatenated vector of $v_{o_1}, v_{o_2}$, and $v_s$ is the concatenated vector of $v_{s_1}, v_{s_2}$. The dimensionality of $v_o$ and $v_s$ are kept the same for a video by adjusting the number of simulation steps we use to obtain $v_o$ according to the length of the video. But from video to video, the length of these vectors may vary. In all of our simulations, we fix $\Sigma$ to 0.05, which is the only free parameter in our model.

## 3.1 Tracking algorithm as a recognition model

The posterior distribution in Equation 1 is intractable. In order to alleviate the burden of posterior inference, we use the output of our recognition model to predict and fix some of the latent variables in the model.

Specifically, we determine the $V_i$, or $\{t_i, x_i, y_i, z_i\}$, using the output of the tracking algorithm, and fix these variables without further sampling them. Furthermore, we fix values of $p_i$s also on the basis of the output of the tracking algorithm.

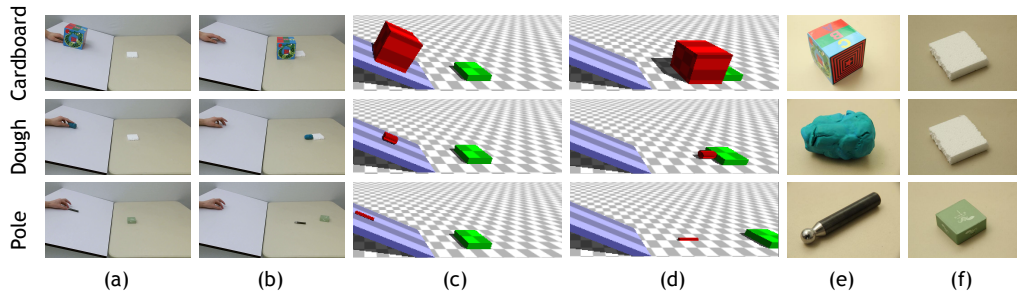

Figure 2: Simulation results. Each row represents one video in the data: (a) the first frame of the video, (b) the last frame of the video, (c) the first frame of the simulated scene generated by Bullet, (d) the last frame of the simulated scene, (e) the estimated object with larger mass, (f) the estimated object with larger friction coefficient.

## 3.2    Inference

Once we initialize and fix the latent variables using the tracking algorithm as our recognition model, we then perform single-site Metropolis Hasting updates on the remaining four latent variables, $m_1, m_2, k_1$ and $k_2$. At each MCMC sweep, we propose a new value for one of these random variables, where the proposal distribution is Uniform$(-0.05, 0.05)$. In order to help with mixing, we also use a broader proposal distribution, Uniform$(-0.5, 0.5)$ at every 20 MCMC sweeps.

## 4    Simulations

For each video, as mentioned earlier, we use the tracking algorithm to initialize and fix the shapes of the objects, $S_1$ and $S_2$, and the position offsets, $p_1$ and $p_2$. We also obtain the velocity vector for each object using the tracking algorithm. We determine the length of the physics engine simulation by the length of the observed video — that is, the simulation runs until it outputs a velocity vector for each object that is as long as the input velocity vector from the tracking algorithm.

As mentioned earlier, we collect 150 videos, uniformly distributed across different object categories. We perform 16 MCMC simulations for a single video, each of which was 75 MCMC sweeps long. We report the results with the highest log-likelihood score across the 16 chains (*i.e.*, the MAP estimate).

In Figure 2, we illustrate the results for three individual videos. Every two frame of the top row shows the first and the last frame of a video, and the bottom row images show the corresponding frames from our model's simulations with the MAP estimate. We quantify different aspects of our model in the following behavioral experiments, where we compare our model against human subjects' judgments. Furthermore, we use the inferences made by our model here on the 150 videos to train a recognition model to arrive at physical object perception in static scenes with the model.

Importantly, note that our model can generalize across a broad range of tasks beyond the ramp scenario. For example, once we infer the density of our object, we can make a buoyancy prediction about it by simulating a scenario in which we drop the object into a liquid. We test some of the generalizations in Section 6.

## 5    Bootstrapping to efficiently see physical objects in static scenes

Based on the estimates we derived from the visual input with a physics engine, we bootstrap from the videos already collected, by labeling them with estimates of Galileo. This is a self-supervised learning algorithm for inferring generic physical properties. As discussed in Section 1, this formulation is also related to the wake/sleep phases in Helmholtz machines, and to the cognitive development of infants.

| Methods | Mass | |
|---|---|---|
| | MSE | Corr |
| Oracle | 0.042 | 0.71 |
| Galileo | 0.052 | 0.44 |
| Uniform | 0.081 | 0 |

Figure 3: Mean squared errors of oracle estimation, our estimation, and uniform estimations of mass on a log-normalized scale, and the correlations between estimations and ground truths

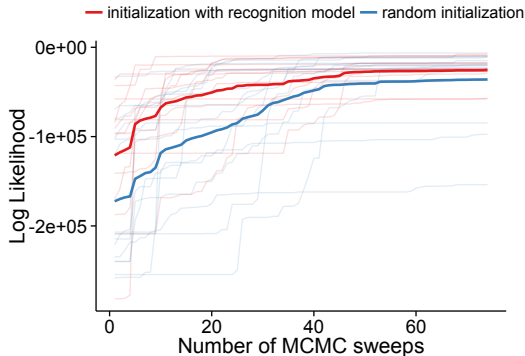

Figure 4: The log-likelihood traces of several chains with and without recognition-model (LeNet) based initializations.

Here we focus on two physical properties: mass and friction coefficient. To do this, we first estimate these physical properties using the method described in earlier sections. Then, we train LeNet [7], a widely used deep neural network for small-scale datasets, using image patches cropped from videos based on the output of the tracker as data, and estimated physical properties as labels. The trained model can then be used to predict these physical properties of objects based on purely visual cues, even though they might have never appeared in the training set.

We also measure masses of all objects in the dataset, which makes it possible for us to quantitatively evaluate the predictions of the deep network. We choose one object per material as our test cases, use all data of those objects as test data, and the others as training data. We compare our model with a baseline, which always outputs a uniform estimate calculated by averaging the masses of all objects in the test data, and with an oracle algorithm, which is a LeNet trained using the same training data, but has access to the ground truth masses of training objects as labels. Apparently, the performance of the oracle model can be viewed as an upper bound of our Galileo system.

Table 3 compares the performance of Galileo, the oracle algorithm, and the baseline. We can observe that Galileo is much better than baseline, although there is still some space for improvement.

Because we trained LeNet using static images to predict physical object properties such as friction and mass ratios, we can use it to *recognize* those attributes in a quick bottom-up pass at the very first frame of the video. To the extent that the trained LeNet is accurate, if we initialize the MCMC chains with these bottom-up predictions, we expect to see an overall boost in our log-likelihood traces. We test by running several chains with and without LeNet-based initializations. Results can be seen in Figure 4. Despite the fact that LeNet is not achieving perfect performance by itself, we indeed get a boost in speed and quality in the inference.

## 6   Experiments

In this section, we conduct experiments from multiple perspectives to evaluate our model. Specifically, we use the model to predict how far objects will move after the collision; whether the object will remain stable in a different scene; and which of the two objects is heavier based on observations of collisions. For every experiment, we also conduct behavioral experiments on Amazon Mechanical Turk so that we may compare the performance of human and machine on these tasks.

### 6.1   Outcome Prediction

In the outcome prediction experiment, our goal is to measure and compare how well human and machines can predict the moving distance of an object if only part of the video can be observed.

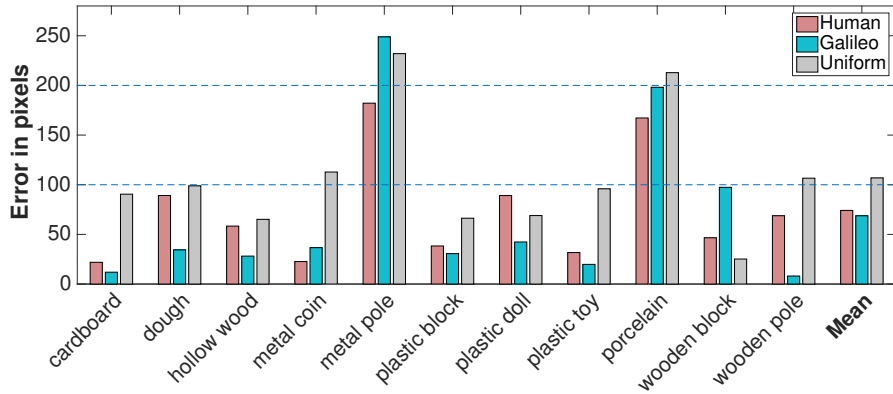

Figure 5: Mean errors in numbers of pixels of human predictions, Galileo outputs, and a uniform estimate calculated by averaging ground truth ending points over all test cases

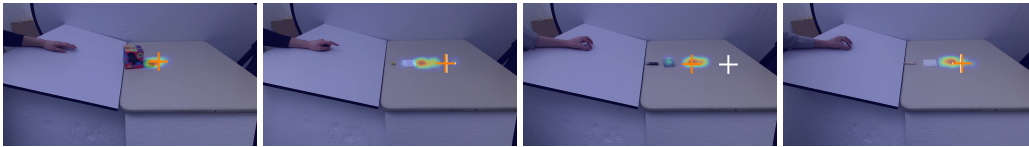

Figure 6: Heat maps of user predictions, Galileo outputs (orange crosses), and ground truths (white crosses).

Specifically, for behavioral experiments on Amazon Mechanical Turk, we first provide users four full videos of objects made of a certain material, which contain complete collisions. In this way, users may infer the physical properties associated with that material in their mind. We select a different object, but made of the same material, show users a video of the object, but only to the moment of collision. We finally ask users to label where they believe the target object (either cardboard or foam) will be after the collision, *i.e.*, how far the target will move. We tested 30 users per case.

Given a partial video, for Galileo to generate predicted destinations, we first run it to fit the part of the video to derive our estimate of its friction coefficient. We then estimate its density by averaging the density values we derived from other objects with that material by observing collisions that they are involved. We further estimate the density (mass) and friction coefficient of the target object by averaging our estimates from other collisions. We now have all required information for the model to predict the ending point of the target after the collision. Note that the information available to Galileo is exactly the same as that available to humans.

We compare three kinds of predictions: human feedback, Galileo output, and, as a baseline, a uniform estimate calculated by averaging ground truth ending points over all test cases. Figure 5 shows the Euclidean distance in pixels between each of them and the ground truth. We can see that human predictions are much better than the uniform estimate, but still far from perfect. Galileo performs similar to human in the average on this task. Figure 6 shows, for some test cases, heat maps of user predictions, Galileo outputs (orange crosses), and ground truths (white crosses).

## 6.2 Mass Prediction

The second experiment is to predict which of two objects is heavier, after observing a video of a collision of them. For this task, we also randomly choose 50 objects, we test each of them on 50 users. For Galileo, we can directly obtain its guess based on the estimates of the masses of the objects.

Figure 7 demonstrates that human and our model achieve about the same accuracy on this task. We also calculate correlations between different outputs. Here, as the relation is highly nonlinear, we

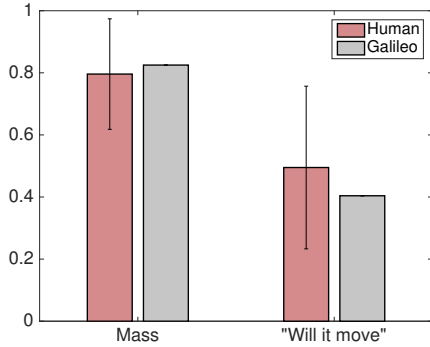

| Mass | Spearman's Coeff |
|---|---|
| Human vs Galileo | 0.51 |
| Human vs Truth | 0.68 |
| Galileo vs Truth | 0.52 |
| | |
| "Will it move" | Pearson's Coeff |
| Human vs Galileo | 0.56 |
| Human vs Truth | 0.42 |
| Galileo vs Truth | 0.20 |

Figure 7: Average accuracy of human predictions and Galileo outputs on the tasks of mass prediction and "will it move" prediction. Error bars indicate standard deviations of human accuracies.

Table 1: Correlations between pairs of outputs in the mass prediction experiment (in Spearman's coefficient) and in the "will it move" prediction experiment (in Pearson's coefficient).

calculate Spearman's coefficients. From Table 1, we notice that human responses, machine outputs, and ground truths are all positively correlated.

### 6.3 "Will it move" prediction in a novel setup

Our third experiment is to predict whether a certain object will move in a different scene, after observing one of its collisions. On Amazon Mechanical Turk, we show users a video containing a collision of two objects. In this video, the angle between the inclined surface and the ground is 20 degrees. We then show users the first frame of a 10-degree video of the same object, and ask them to predict whether the object will slide down the surface in this case. We randomly choose 50 objects for the experiment, and divide them into lists of 10 objects per user, and get each of the item tested on 50 users overall.

For Galileo, it is straightforward to predict the stability of an object in the 10-degree case using estimates from the 20-degree video. Interestingly, both humans and the model are at chance on this task (Figure 7), and their responses are reasonably correlated (Table 1). Moreover, both subjects and the model show a bias towards saying "it will move." Future controlled experimentation and simulations will investigate what underlies this correspondence.

## 7 Conclusion

This paper accomplishes three goals: first, it shows that a generative vision system with physical object representations and a realistic 3D physics engine at its core can efficiently deal with real-world data when proper recognition models and feature spaces are used. Second, it shows that humans' intuitions about physical outcomes are often accurate, and our model largely captures these intuitions — but crucially, humans and the model make similar errors. Lastly, the experience of the model, that is, the inferences it makes on the basis of dynamical visual scenes, can be used to train a deep learning model, which leads to more efficient inference and to the ability to see physical properties in the static images. Our study points towards an account of human vision with generative physical knowledge at its core, and various recognition models as helpers to induce efficient inference.

## Acknowledgements

This work was supported by NSF Robust Intelligence 1212849 Reconstructive Recognition and the Center for Brains, Minds, and Machines (funded by NSF STC award CCF-1231216).

## Footnotes

[1]For shape type box, $x_i, y_i$, and $z_i$ could all be different values; for shape type torus, we constrained the scaling factors such that $x_i = z_i$; and for shape type cylinder, we constrained the scaling factors such that $y_i = z_i$.

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
