[Reviews · NeurIPS 2015]

Submitted by Assigned_Reviewer_1

Summary: The authors introduce a novel approach for inferring hidden physical properties of objects (mass and friction), which also allows the system to make subsequent predictions that depend on these properties. They use a black-box generative model (a physics simulator), to perform sampling-based inference, and leverage a tracking algorithm to transform the data into more suitable latent variables (and reduce its dimensionality) as well as a deep model to improve the sampler. The authors assume priors over the hidden physical properties, and make point estimates of the geometry and velocities of objects using a tracking algorithm, which comprise a full specification of the scene that can be input to a physics engine to generate simulated velocities. These simulated velocities then support inference of the hidden properties within an MCMC sampler: the properties' values are proposed and their consequent simulated velocities are generated, which are then scored against the estimated velocities, similar to ABC. A deep network can be trained as a recognition model, from the inferences of the generative model, and also from the Physics 101 dataset directly. Its predictions of the mass and friction can be used to initialize the MCMC sampler. The authors compare performance to a convnet and to various human judgments.

Quality: Good. The approach is well-executed, and provides good performance on an interesting task.

Clarity: The writing is very clear. Few (if any) typos, easy to understand.

Originality: I'm not aware of any competing approach to this problem, and the authors implement a convnet ("oracle" model) to compare their system against.

Significance: This work is strong, and should be of interest to both machine learning and computer vision.

Suggestions: - The outcome prediction analysis (pgs 6-7) should include another comparison besides the error bar graph, such as correlations between human, POM, and uniform, because the summary "Mean" bars in Figure 5 give a misleading impression that human and POM are very similar, despite their patterns of errors being fairly different. - Section 4 says that after inferring physical properties, the system can be applied to novel settings, such as predicting buoyancy, which the authors say will be demoed in Section 6. There is no more talk of buoyancy, however, so this statement should be removed.
Summary: The paper should be accepted because it offers a novel amalgam of deep learning and rich generative models to tackle a complex, real-world physical reasoning problem. The writing is clear, the technical work is well-executed, and the performance is thoroughly tested in various ways.

Submitted by Assigned_Reviewer_2

This is a high quality paper, clearly written, original, and has the potential for considerable significance.

As I see it, the major contribution and innovation here is that the authors connect a symbolic physical model to an image-level recognition and tracking algorithm.

As such, I think the authors need to dedicate a bit more space to discussing the image-level recognition/tracking model, its capabilities and limitations.

The scenarios they assess are rather impoverished images; how well would this model fair with new scene configurations? with surfaces that are not smooth planes?

With objects that are not blocks?

Insofar as this is indeed as general an algorithm as it might seem at first blush, this is worth advertising clearly.

If it is limited to fairly simple scenes with known geometric configurations (which would be an easy way to make something that operates over image data, but only because the domain of objects is very impoverished), this needs to be clearly stated.

Either way, this reflects useful progress, but the former scenario is far more impressive.

A few other comments: - In the introduction (lines ~96-98) the authors make the point that a "computational solution asserts that the infant starts with a relatively reliable mental physics engine, or acquires it soon after birth". This statement sounds like a nativist claim that I don't believe the authors would hold fast to - if infants developed a mature physics engine throughout the first year or two of life, the findings from this paper would still hold (and I would not count this as 'soon after birth'). In fact, much of Renee Baillaregon's work demonstrates that physical reasoning performance develops throughout the first year of life. If the authors wish to claim that mental physics engines are developed (and well formed) very early, they should give a nod to this work; however, I believe this statement distracts from the overall message of the paper and could be dropped.

- On line 198 the authors fix sigma (the velocity noise) to 0.05. However, it is not clear how this value was set, or how sensitive to changes in this parameter the model is. It would be helpful to provide units for this value (is it in m/s? normalized to average velocity?) and briefly demonstrate that the model predictions do not change significantly under a couple other reasonable values for this parameter.

- In the Outcome Prediction experiment (6.1), it is mentioned that the target object is either cardboard or foam, but the x-axis in Fig 5 represents only (I believe) the material of the initially moving object. Are the errors in Fig 5 averaged across both target objects, or was each initial material associated with only one of the two target objects?

- It seems useful to estimate error correlations between people, the uniform model, and POM on the data in figure 5, given the wide variation in errors across all scenes.

- For the other experiments (Mass Prediction & "Will It Move") it's not clear how the data is being analyzed. In both cases, I believe the human decision is dichotomous (obj1 heavier vs obj2 heavier for Mass Prediction and moving vs not for Will it Move), and assume that this is aggregated for each trial to get the proportion of people making each decision. Then in both cases, the model samples from the posterior a number of times to get a proportion of either choice for each model? Further clarity on this analysis would help readers understand these results

- In the Mass Prediction experiment, the correlations between human and model data are calculated using Spearman's rho because the relationship is claimed to be very non-linear. However, if the POM is capturing human judgments accurately then there should be a linear, 1:1 relationship. Therefore, Pearson's correlation would be more appropriate for this relationship (unless the correlation would be skewed by outliers). Showing scatterplots of these relationships by trial would also help readers understand this relationship.

- In the first sentence of the paper (line 37) it is a *Rube* Goldberg machine
Summary: The paper introduces a physical reasoning system that combines a physics engine and an image-level tracking algorithm to infer latent physical properties of objects, and evaluate predictions of this system against humans in a number of experiments.

This work is a useful contribution that has the potential to go far beyond current physical reasoning models by virtue of dealing directly with image-level data.

Submitted by Assigned_Reviewer_3

This paper takes on the question of whether it is possible to make predictions about the temporal evolution of objects in a scene, from watching video streams of that scene. For instance, when we see a block move a certain way, we form an expectation of where it is going, presumably because of an intuitive physics model that is part of our cognitive system. This subject has been explored in recent literature, some of which is cited by the authors.

The contribution of this paper is to utilise a deep learning tool to make predictions from visual features, which is coupled with a Bayesian generative formulation of the dynamics of the objects. The Bayesian formula in eq 1 is a standard statement of how physical parameters get translated into visual features. The authors take the observed velocity, via a tracker, as the observable of interest.

The main contribution of the paper, as I understand it, is to utilise these tools to conduct an experiment wherein the system is shown to be capable of making predictions about the evolution of the scene based on partial traces of the video. We see many variations on this experiment, to show both that the system can 'classify', i.e., make predictions about discrete labels such will or will not hit a point, as well as continuous variables such as how much motion we will see.

If I were to look beyond the fact that LeNet allows one to get at visual features, the rest of the argument is not all that surprising. To the extent that the simulator acts as parameterised model, some of whose parameters are unknown at the start, what the authors are doing is exactly the same thing as inverse problems that have long been solved by meteorologists and oil industry professionals who do this with much more complex forms of dynamics (e.g., fluid flow and the estimation of viscosity). So, is it is really surprising that eq 1 can be used to estimate mass of the object?

The comparison to human perception is interesting and not something that statisticians who look at inverse problems have considered before. However, we don't get much elaboration of this. Indeed, the sprinkling of cognitive hypotheses throughout the paper is of a very cursory kind without substantive claims and argument. For instance, in pp 2, we have the claim that a certain type of model would need a good physics engine to be innate - a big assumption one way or the other. However, that paragraph is left hanging without further discussion.

Ultimately, beyond the timeliness of the use of a deep learning tool for getting at visual features (not something that is being advanced as an innovation within this paper), I am unsure of the novel and significant contributions of this paper.
Summary: This paper uses a generative model formulation to infer physical properties of objects in a video dataset, to show that useful predictions can be made about future evolution of the scene. The generative model itself is a standard Bayesian formula, which is coupled with a deep learning tool to make predictions from the image(s). While the results seem valid, this reviewer is unsure of its significance above and beyond what is already known from the prior work on the same topic.

Author Feedback
Author rebuttal: We thank reviewers for their comments and time. We appreciate the positive comments, and also the points raised.

[Model]
R2: the image-level recognition/tracking model and its general applicability
In our current system, we are using a KLT tracker, a classic computer vision algorithm for tracking. On the Physics 101 dataset, the tracker performs well. In a more general case (e.g. complex background, multiple objects), a KLT tracker might not be sufficient; but note that there have been many more involved techniques for such cases (e.g. [A]). Given our system is independent of the tracking model, we could simply replace the KLT tracker with them if necessary. Regarding geometric configurations, the current model can deal with shape categories including tori, cylinders, and boxes. A potential future study is to incorporate 3d reconstruction pipelines to broaden the space of object shapes. We will include these discussions into the paper.

[A] Wang and Yeung. Learning a deep compact image representation for visual tracking. NIPS 2013.

[Analysis]
R2: Analysis of Mass Prediction & "Will It Move" experiments
As R2 mentioned, for both experiments human responses were aggregated for each trial to get the proportion of people making each decision. In order to evaluate the performance of POM, we use the MAP estimate (i.e., the sample with the highest posterior score) as the outcome for each trial. For correlation analyses, we use the ratio of the mass 1 and mass 2 estimated by POM for the Mass Prediction experiment, and the friction coefficient of the source object in the "Will it Move" task.

R2: Justification for Spearman's rho in calculating correlations
Because we use the mass ratio of the two objects as our predictor, we are trying to map a range of values from ~0 to ~infinity to the range 0 and 1 (average subject responses), through a potential sigmoidal non-linearity (e.g., when the value of the predictor is 1, the expected behavioral value is 0.5, which should decrease to 0 quickly for predictor values smaller than 1, and should increase to 1 for predictor values greater than 1).

R1 & R2: Correlations between human, POM, and uniform for outcome prediction
The error correlation between human and POM is 0.70. Unfortunately, the correlation analysis for the uniform model is not useful: independently of the choice of the exact value of the uniform prediction (i.e., be it the actual average of all the trials or just a randomly chosen value such as 0), the correlation is high and constant (about 0.72 for both human and POM predictions). We will include this analysis, as well as a scatter plot if space permits, in the revised version.

[Novelty]
R3: Contributions
Our main contribution in this paper, as suggested by R1, R2, and R4, is an integrated system combining the advantages of discriminative features extracted via deep learning and a symbolic generative physical model for the goal of real-world physical scene understanding. Our model formulates and provides an algorithm to solve a physical inverse problem. Differently from modeling complex fluid dynamics predictions, we explore our model as a cognitive theory of human perception by comparing human observers' responses and the model's predictions, finding that the two correspond fairly. We will make the cognitive arguments clearer in the paper.

[Others]
R1: Buoyancy
We will replace the statement about buoyancy with one about the experiments in Section 6. The point we would like to make is that the generative nature of the model ensures its extensive generalization capacity.

R2, R3: Infants innate knowledge of physics engine
As R2 suggested, the claim itself is not crucial, and we will drop it in the revised version.

R2: sigma fixed to 0.05
Yes, the unit of this parameter is m/s. We tried various values for sigma (smallest value being 0.01) during the development of the model, which did not change our results in a qualitative manner.

R2: Target objects in Fig 5
The errors in Fig 5 are averaged across both target objects for each material. The error patterns are similar for both kinds of target objects (foam and cardboard).

R4: paper includes too much, not enough details
We will provide more details in the supplementary material.

R5: Related work [1]
The only overlap between this submission and [1] is that we used a subset of the Physics 101 dataset proposed in [1]. What were introduced in this paper, including the POM model, and all simulations and experiments, are essentially different from those in [1].

R6: Best human performance
Each instance is tested on 30 to 50 users; but we did not require any user to complete all of the trials. Actually, some of the users only performed one trial. Therefore, we cannot evaluate how well an individual participant can perform our full set of trials.

We will also revise other minor points raised by the reviewers.